# Evaluation of 3D Templated Synthetic Vascular Graft Compared with Standard Graft in a Rat Model: Potential Use as an Artificial Vascular Graft in Cardiovascular Disease

**DOI:** 10.3390/ma14051239

**Published:** 2021-03-05

**Authors:** Sung-Hwa Sohn, Tae-Hee Kim, Tae-Sik Kim, Too-Jae Min, Ju-Han Lee, Sung-Mook Yoo, Ji-Won Kim, Ji-Eun Lee, Chae-Hwa Kim, Suk-Hee Park, Won-Min Jo

**Affiliations:** 1Department of Thoracic & Cardiovascular Surgery, Korea University Ansan Hospital, Korea University College of Medicine, Ansan 15355, Korea; iisupy@korea.ac.kr (S.-H.S.); kmdphd@gmail.com (T.-S.K.); sungmook84@korea.ac.kr (S.-M.Y.); 2Advanced Textile R&D Department, Korea Institute of Industrial Technology, Ansan 15588, Korea; thkim75@kitech.re.kr (T.-H.K.); jieun919@kitech.re.kr (J.-E.L.); purech@kitech.re.kr (C.-H.K.); 3Department of Anesthesiology and Pain Medicine, Korea University Ansan Hospital, Korea University College of Medicine, Ansan 15355, Korea; minware@korea.ac.kr (T.-J.M.); sient0825@nate.com (J.-W.K.); 4Department of Pathology, Korea University Ansan Hospital, Korea University College of Medicine, Ansan 15355, Korea; repath@korea.ac.kr; 5School of Mechanical Engineering, Pusan National University, 2 Busandaehak-ro, 63 Beon-gil, Geumjeong-gu, Busan 46241, Korea

**Keywords:** 3D templated graft, thrombogenesis, calcification, inflammation

## Abstract

Although the number of vascular surgeries using vascular grafts is increasing, they are limited by vascular graft-related complications and size discrepancy. Current efforts to develop the ideal synthetic vascular graft for clinical application using tissue engineering or 3D printing are far from satisfactory. Therefore, we aimed to re-design the vascular graft with modified materials and 3D printing techniques and also demonstrated the improved applications of our new vascular graft clinically. We designed the 3D printed polyvinyl alcohol (PVA) templates according to the vessel size and shape, and these were dip-coated with salt-suspended thermoplastic polyurethane (TPU). Next, the core template was removed to obtain a customized porous TPU graft. The mechanical testing and cytotoxicity studies of the new synthetic 3D templated vascular grafts (3DT) were more appropriate compared with commercially available polytetrafluoroethylene (PTFE) grafts (ePTFE; standard graft, SG) for clinical use. Finally, we performed implantation of the 3DTs and SGs into the rat abdominal aorta as a patch technique. Four groups of the animal model (SG_7 days, SG_30 days, 3DT_7 days, and 3DT_30 days) were enrolled in this study. The abdominal aorta was surgically opened and sutured with SG or 3DT with 8/0 Prolene. The degree of endothelial cell activation, neovascularization, thrombus formation, calcification, inflammatory infiltrates, and fibrosis were analyzed histopathologically. There was significantly decreased thrombogenesis in the group treated with the 3DT for 30 days compared with the group treated with the SG for 7 and 30 days, and the 3DT for 7 days. In addition, the group treated with the 3DT for 30 days may also have shown increased postoperative endothelialization in the early stages. In conclusion, this study suggests the possibility of using the 3DT as an SG substitute in vascular surgery.

## 1. Introduction

In 2016, cardiovascular disease (CVD) led to 1.68 million deaths in the European Union [1]. Although endovascular treatment for vascular disease is popular and considered the first option for treatment in many cases, the size of vascular graft market has increased continuously and rapidly [2,3]. In other words, the management of diseases such as coronary and peripheral artery disease, congenital heart defects, and end-stage renal disease often requires vascular surgery for revascularization or formation of a fistula track using grafts of various types and sizes [4]. The safety and appropriateness of a vascular graft determine the treatment outcomes in CVD, which is the leading cause of death. In addition to vascular graft-related complications such as infection, thrombogenesis, rupture, and (pseudo-)aneurysmal changes, the lack of availability of small-diameter (<6 mm) vascular grafts is another challenge limiting the use of vascular grafts clinically due to a high risk of intimal hyperplasia, luminal thrombosis, inflammation, and consequent compliance mismatch with human vessels [5,6].

Commercial vascular grafts (polytetrafluoroethylene, ePTFE; standard graft, SG) have been frequently used in clinical applications. However, the SG has low efficacy relative to an autologous and tissue-engineered vascular graft because of the slow rate of endothelialization and increased risk of thrombogenesis. Endothelialization is a crucial factor for long-term implantation due to its outstanding anticoagulant effects, resulting in resistance to inflammation and thrombosis [7,8,9,10]. Endothelialization of SGs is slow and difficult because of the highly crystalline and hydrophobic behavior toward endothelial cell adhesion, spread, and growth [11,12], therefore, SGs lack luminal surface endothelial cell coverage after graft implantation in human vessel. In addition, SGs show poor long-term patency because of platelet adhesion and the adsorption of plasma proteins, which may induce early thrombus formation [7,8]. Moreover, the porosity of vascular grafts is positively correlated with endothelialization and vascularization [13]. However, materials with standard low-porosity SGs (internodal distance ≤ 30 μm) could be enriched with amorphous platelets or eventually a thin fibrin coagulum after human implantation [9,14]. To overcome these limitations, the current study explored the strategies to effectively promote endothelialization and inhibit thrombogenicity for developing improved vascular grafts compared with commercially available grafts in addition to appropriate mechanical and cytological properties.

Therefore, we re-designed the vascular graft with modified materials and 3D printing techniques and also demonstrated the improved clinical applications of our new vascular graft. First, we developed the 3D templated vascular graft (3DT), followed by mechanical testing and cytotoxicity studies comparing 3DTs and SGs. Rats and humans exhibit similar blood pressure and homeostatic mechanisms [15]. Later, we performed implantation of 3DTs and SGs into the rat abdominal aorta using a patch technique. The purpose of our study is to investigate the clinical efficacy of 3DTs compared with SGs.

## 2. Materials and Methods

### 2.1. Fabrication of Artificial Vascular Graft

The original techniques for the fabrication of artificial vascular grafts have been reported previously in detail [16]. In brief, polyvinyl alcohol (PVA) filament feedstocks (ESUN, Shenzhen, China) were processed into sacrificial templates for the shaping of grafts using a material extrusion 3D printer (Ender 3 pro K, Creality, China) equipped with a nozzle measuring 0.4 mm in inner diameter. The printing conditions were set at a nozzle temperature of 200 °C and a nozzle speed of 60 mm/s. The 3D printed PVA templates were dip-coated with a salt-suspended thermoplastic polyurethane (TPU) solution, which was prepared by mixing salt powders (MW 0.44, Sigma Aldrich, St. Louis, MO, USA) and a TPU solution at a ratio of 4:1 (*w/w*). The TPU (DAELIM chemical, Korea) granules were dissolved in dimethylformamide (DMF, Samchun Pure Chemical, Korea) at a concentration of 15% (*w/v*) to obtain the solution. After dipping the PVA template into the TPU–salt suspension, the dip-coated template was dried and thereafter immersed in sonicated water to leach out the salt powders. In the final step, the core template was removed through cut ends in the sonicated water, resulting in a customized porous TPU graft.

### 2.2. Mechanical Characterization of Vascular Grafts

The mechanical properties of customized graft were evaluated and compared with those of an SG. Uniaxial tensile tests were performed using a typical universal testing machine (Instron 3367, Norwood, MA, USA). Tensile specimens were cropped from the as-prepared artificial graft measuring 15 mm in diameter and 45 mm in length. The graft was cut into rectangular sheet pieces with a width of 6 mm and a length of 40 mm. Using the resized samples, the tensile tests were performed at a rate of 500 mm/min and a gauge length of 20 mm. In total, five samples were tested for the measurement of average elongation at break and elastic modulus (20% secant).

### 2.3. Surface Characterization of Vascular Grafts

The porous structure of 3DT and SG inner surfaces was analyzed via field emission scanning electron microscopy (FE-SEM, Hitachi SU-8010, Hitachi High-Tech Co., Tokyo, Japan). Prior to imaging, the samples were sputter-coated with gold for 250 s using a 15 mA current.

### 2.4. In Vitro Cell Studies

#### 2.4.1. Cell Culture

L-929 mouse fibroblast cells obtained from the Korean Cell Line Bank (KCLB, Seoul, Korea) were cultured in RPMI 1640 medium (Hyclone) supplemented with 10% fetal bovine serum (FBS, Hyclone) and 1% penicillin/streptomycin at 37 °C in a humidified 5% CO_2_ environment.

#### 2.4.2. Cytotoxicity

Cell viability was evaluated based on ISO 10993-5 using a CellTiter 96^®^ AQueous Cell Proliferation Assay kit (Promega Corp., Madison, WI, USA). Extracts of sterilized grafts (3DT, SG) were obtained by placing in RPMI 1640 medium at an extraction ratio of 0.1 g/mL at 37 °C, 100 rpm for 24 h. L-929 cells were seeded in a 96-well culture plate at a density of 1 × 10^4^ cells/well and incubated for 24 h. The culture medium was replaced with graft extracts and incubated for an additional 48 h. Cell morphology in each well was observed with an inverted light microscope (Leica, Wetzlar, Germany). Then, the medium was replaced with the medium containing MTS (3-(4,5-dimethylthiazol-2-yl)-5-(3-carboxymethoxyphenyl)-2-(4- sulfophenyl)-2H-tetrazolium) reagent and incubated for 3 h. Absorbance was measured with a microplate reader (SpectraMax iD3, Molecular Device Co, Ltd, San Jose, CA, USA) at a wavelength of 490 nm. The cell viability (%) was calculated using the following equation:Cell viability (%) = OD_490_ (sample) / OD_490_ (control) × 100(1)

### 2.5. In Vivo Animal Studies

#### 2.5.1. Animals

Sixteen-week-old specific pathogen-free Sprague Dawley male rats (total number = 24, 440~460 g) were used in this study. All of animal housing, breeding, and experiments were approved by the Animal Care and Use Committee of Korea University (KOREA-2019-0060-C1).

#### 2.5.2. Experimental Design

The operation was done according to our previously described method [16]. On day 0, rats were anesthetized with isoflurane. A longitudinal laparotomy incision was made and the infrarenal abdominal aorta was exposed. After 3 min of heparin injection (50 IU/kg, intra-peritoneally), the abdominal aorta was cross-clamped and opened longitudinally to a length of 1.5 cm. The opened aorta was sutured and covered with an SG or 3DT vascular patch. The wound was closed layer by layer (Figure 1a–h). In total, four groups of the animal model were enrolled in this study: SG_7 day (standard graft; ePTFE), SG_30 day, 3DT_7 day (customized vascular patch; 3D templated graft), and 3DT_30 day. After 7 to 30 days, the aorta and surrounding tissues of these operated animals were harvested for histopathologic analysis after euthanasia. Figure 1i shows the experimental design.

#### 2.5.3. Tissue Harvesting and Staining

After extraction of the aorta and surrounding tissues, these samples were fixed with 4% paraformaldehyde and stained with hematoxylin and eosin (H&E). After that, histopathological analysis including the luminal thrombus, neovascularization, calcification, fibrosis, and inflammatory infiltrates was conducted.

#### 2.5.4. Histopathology Scoring

Histopathology grading of thrombi, calcification, neovascularization, inflammatory cell infiltrates, and fibrosis was evaluated and rated by a pathologist in a blinded manner. This grading consisted of grade 0 (for none), 1 (for mild), 2 (for moderate), and 3 (for severe) and scoring was calculated as a mean value.

### 2.6. Statistical Analysis

The data are expressed as the mean ± standard deviation (SD). Statistical significance was determined using the Mann–Whitney test. A *p*-value of <0.05 is considered to indicate statistical significance. All statistical analyses were conducted using GraphPad Prism software (GraphPad, La Jolla, CA, USA).

## 3. Results

### 3.1. 3D Customized Artificial Vascular Grafts

We previously reported a template-based fabrication of tubular tissue engineering scaffolds [17,18]. In this study, using the same process, the artificial vascular grafts were fabricated and customized using a 3D printing technique. Figure 2a shows the 3D printing of a cylindrical PVA template. Artificial grafts with morphology similar to the corresponding templates were obtained after dip-coating in the TPU–salt suspension, salt leaching, and the removal of the PVA template. Owing to the inherent customizability of the 3D printing method, the artificial grafts could be custom-made to clinical specifications. Figure 2b displays two different grafts with inner diameters of 5 mm (upper) and 10 mm (lower). In addition to facile dimensional control, the artificial grafts exhibit good mechanical flexibility for clinical efficacy, as shown in Figure 2c.

To evaluate the mechanical flexibility quantitatively, tensile tests were conducted using the as-prepared sheet specimens cut from the artificial grafts. Figure 3a,b show the resulting stress–strain relationship of specimens cropped from the as-fabricated porous 3DT. Compared with the SG with elongation at break of 89% and elastic modulus of 61 MPa (25% secant), the porous 3DT exhibited superior flexibility with 456% elongation and an elastic modulus of 19 MPa (Figure 3a,b). The advantageous properties were attributed to the intrinsic softness of the material (TPU) and the high porosity generated by salt leaching. Figure 3c,d show the porous structures of the inner wall surfaces of the 3DT and the SG samples. A higher porosity and larger pore sizes were observed in the 3DT sample compared to those of the SG sample. The similar tendency was revealed in the cross-sectional views of the samples. As expected, the differentiated pore morphology of 3DT caused the superior mechanical flexibility over that of SG.

### 3.2. In Vitro Cytotoxicity

Next, the viability of L-929 cells with or without extracts of 3DT or SG was investigated using an MTS assay. As shown in Figure 4, the cell viability of the 3DT or SG was around 90% and 106%, respectively.

### 3.3. In Vivo Implantation

#### 3.3.1. Endothelialization

To evaluate the postoperative effects of the 3DT vascular patch, the endothelial cell activation was compared with that of the SG for 7 or 30 days postoperation. The endothelial cells are originally flat cells, but when activated they become round. The SG and 3DT groups revealed endothelial cell activation on day 7 (Figure 5), whereas no endothelial cell activation was detected in both groups on day 30. Endothelialization is a crucial factor for long-term implantation due to anticoagulation, which decreases thrombus formation and prolongs implant function [7,19]. The 3DT_7 day group revealed higher endothelial activation compared with the SG_7 day group (Figure 5). These data reveal that endothelialization indicates activation at an early stage, but is not continued in both vascular patches.

#### 3.3.2. Histopathological Analysis

To further investigate the postoperative effects of the 3DT vascular patch, the histopathological findings of neovascularization, luminal thrombus, calcification, inflammatory cell infiltrates, and fibrosis were compared with those of the SG for 7 or 30 days postoperation (Figure 6). Among the groups, the SG_30 day group manifested slightly higher neovascularization (Figure 6a,e). The 3DT_30 day group revealed slightly higher fibrosis compared with the SG_30 day group (Figure 6e). However, the SG_30 day group showed advanced fibrosis, whereas the 3DT_30 day group exhibited early fibrosis (Figure 6b,d). The SG_30 day and 3DT_30 day groups revealed higher calcification compared with the SG_7 day and 3DT_7 day groups (Figure 6e). Calcification of blood vessels occurs along with normal aging but otherwise represents the initiation of thrombi [20,21]. The SG_30 day group revealed slightly higher calcification compared to the 3DT_30 day group (Figure 6e). In addition, among these four groups, only the 3DT_30 day group had significantly less thrombus formation (Figure 6e). However, neovascularization, inflammation, and fibrosis were not significantly different in all groups.

## 4. Discussion

The ideal vascular graft should exhibit appropriate mechanical strength and compressibility, permeability and viscoelasticity, biocompatibility and biostability, hemocompatibility and non-thrombogenicity. These prerequisites have already been reported in many studies [22,23,24,25]. In addition, the ideal graft meets the physiological, manufacturing and optional needs. Physiologically, the ideal vascular graft cannot disturb tissue healing, and should be non-toxic. It should also be devoid of antigenicity, oncogenicity, and adverse immune effects, and suppress intimal hyperplasia [26,27,28,29,30]. Manufacturing conditions for the ideal vascular graft should be easy and accurately reproducible. Such grafts can be rapidly manufactured with diverse diameters and lengths as well as shapes, and easily stored and shipped. In this respect, our method, including 3D printing and solution coating, is characterized with facile manufacturing and shape customizability. Besides, drug elution is often required clinically. The manufacturing price is one of the most important prerequisites [31,32,33,34,35]. The prevalence of vascular disease is increasing rapidly because of the graying of the population. In addition, morbidity and mortality related to vascular ailments are also increasing. In fact, vascular disorders can deteriorate the quality of life in patients and entail enormous social and healthcare costs. Therefore, many investigations and studies have focused on elimination of vascular disease for several decades. These efforts have resulted in significant achievements, especially in endovascular treatment. However, the role of vascular surgery using the vascular graft with or without concomitant endovascular treatment is still important clinically. Thus, many of the challenges and limitations related to vascular grafts, such as infection, thrombus formation, rupture, and (pseudo-)aneurysmal changes, need to be resolved. Accordingly, more advanced vascular grafts with a diverse array of materials and manufacturing techniques are necessary. In addition, these grafts should be commercially available “off the shelf”.

This study introduces a new synthetic 3DT. Although this study had some limitations due to the small sample size and animal operations using a patch technique, which is not a full graft technique, it suggests the possibility of clinical applications using the newly fabricated synthetic 3DT. Our synthetic graft was not inferior compared to the commercially available SG, which is used worldwide clinically, in terms of mechanical strength, surface morphology, cytotoxicity, and histological results following in vivo implantation. The elastic moduli of typical vascular tissues range from approximately 1 MPa to 40 MPa, depending on the type of artery or vein [36,37,38]. In this context, the SG sample showed excessively high stiffness, exceeding the feasible range of native vascular tissues. In contrast, our graft is soft and alleviates the mechanical mismatch between graft and native tissue. The 3DT graft is expected to provide great potential as a biomimetic vascular graft owing to its mechanical softness, which has been continuously challenged in previous studies [17,39,40]. The cytotoxicity of our synthetic 3DT is not different from that of the commercially available SG. In addition, significantly reduced calcification and thrombus formation were observed in the 3DT_30 day group compared with the SG_30 day group in our study. Fibrosis and thrombus formation may induce structural degeneration, including calcification [41], and vascular calcification is generally seen with aging [20]. Of course, this study did not establish the superiority of the newly fabricated graft compared to the commercially available SG. However, this study represents a preliminary analysis to determine the possibility of clinical applications using the synthetic 3DT. In addition, our graft has advantages related to an additional layer of coating with repurposed materials and is designed to improve clinical outcomes, such as reduced risk of thrombogenesis and intimal hyperplasia via drug-eluting techniques, and facilitates endothelial cell activation by modifying the coating of the graft’s inner layer. Our new technique facilitates the fabrication of patient-specific synthetic grafts tailored to individual patients based on data obtained using computed tomography or magnetic resonance imaging, especially in diverse branched and severely distorted vessels. We are currently investigating the functionally improved and patient-specific customized 3DT and will report the detailed results soon.

## 5. Conclusions

Our study revealed that the 3DT was biocompatible and reduced the risk of thrombogenesis in the early postoperative stages of vascular surgery compared with the SG. These results suggest the possibility of clinical applications using the 3DT as a substitute for pre-existing SGs in vascular surgeries.

## Figures and Tables

**Figure 1 materials-14-01239-f001:**
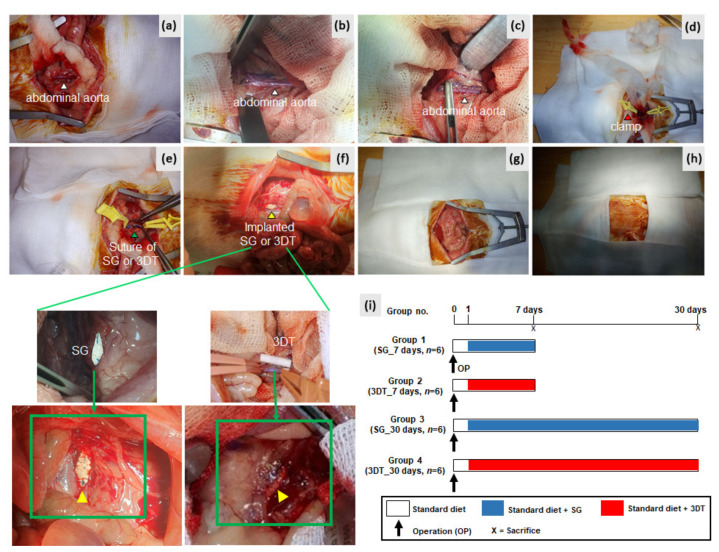
In vivo implantation of standard graft (SG) or 3D templated vascular graft (3DT) vascular patches using a rat abdominal aorta model. (**a**–**c**) White arrows indicate abdominal aorta; (**d**), red arrow indicates clamp; (**e**) green arrow indicates suture with SG or 3DT vascular patch; (**f**), yellow arrow indicates implanted SG or 3DT vascular patch; (**g**,**h**) the wound was closed layer by layer; (**i**) experimental design.

**Figure 2 materials-14-01239-f002:**
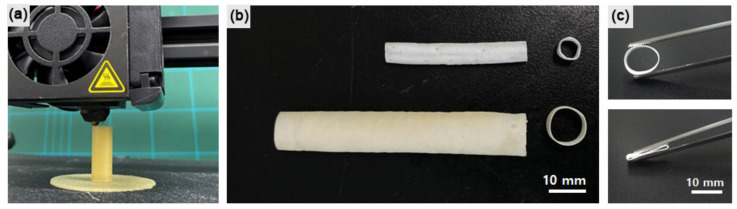
(**a**) 3D printing of sacrificial polyvinyl alcohol (PVA) template. (**b**) Cylindrical artificial vascular grafts. (**c**) Flexibility of the as-fabricated graft.

**Figure 3 materials-14-01239-f003:**
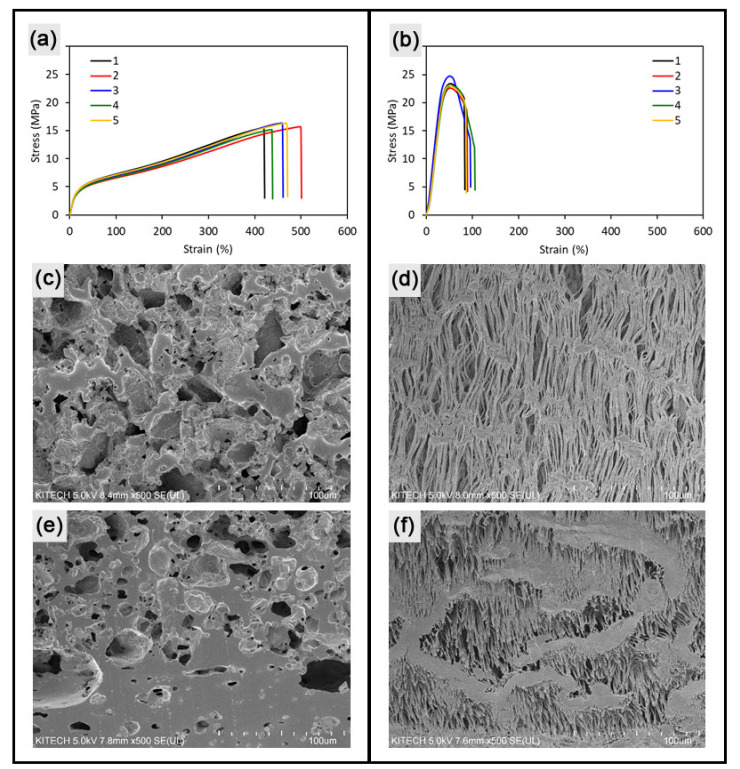
Flexibility of the fabricated graft (3DT) and the SG. Stress–strain curves of tensile specimens cropped from (**a**) the porous 3DT and (**b**) the SG (*n* = 5). SEM images of inner surface of (**c**) 3DT and (**d**) SG. SEM images of cross-section of (**e**) 3DT and (**f**) SG.

**Figure 4 materials-14-01239-f004:**
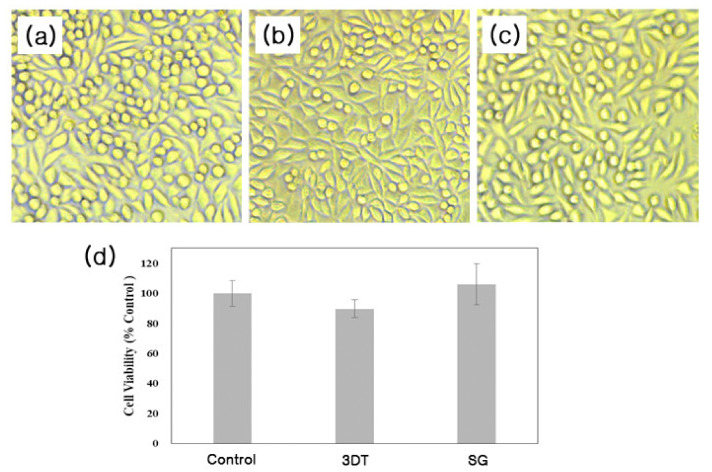
Viability of L-929 cells. Morphology of L-929 fibroblast cells treated with or without 3DT or SG extracts after 48 h incubation. (**a**) Control. (**b**) Customized synthetic 3D templated grafts (3DT). (**c**) SG, standard graft, ePTFE (expanded polytetrafluoroethylene). (**d**) Evaluation of cell viability of 3DT and SG.

**Figure 5 materials-14-01239-f005:**
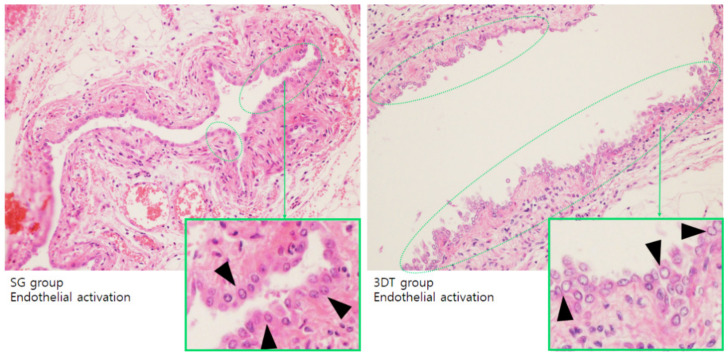
Endothelial activation in SG and 3DT groups. Representative images of hematoxylin and eosin (H&E)-stained aorta and aorta surrounding tissue 7 days postoperation. Black arrows and green dotted lines indicate endothelial activation (magnification, 100×).

**Figure 6 materials-14-01239-f006:**
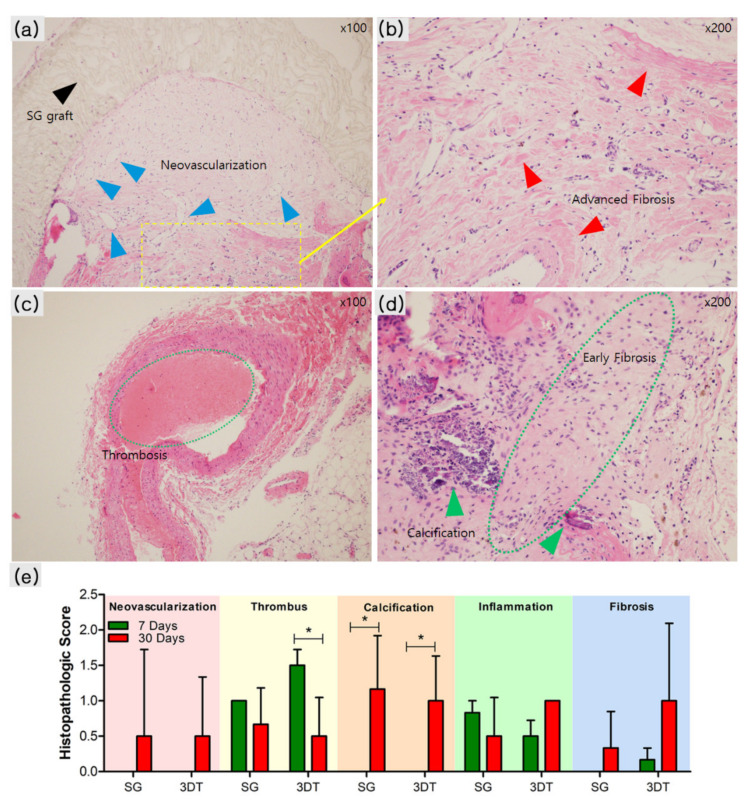
Histopathological observations in (**a**,**b**) SG and (**c**,**d**) 3DT groups. (**a**–**d**) Representative images of H&E-stained aorta and surrounding tissue 30 days postoperation. Black arrow indicates SG graft: standard graft, ePTFE. Blue arrows indicate neovascularization. Red arrows denote advanced fibrosis. Small green dotted lines represent thrombosis; large green dotted lines suggest early fibrosis; green arrows indicate calcification. 3DT: 3D templated graft (magnification, 100× and 200×). (**e**) Histopathological score in SG and 3DT groups. Data are expressed as the mean ± standard deviation. * *p* < 0.05 compared with 7-day group.

## Data Availability

The data presented in this study are available on request from the corresponding author.

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
