# Peer review of "Evaluation of 3D Templated Synthetic Vascular Graft Compared with Standard Graft in a Rat Model: Potential Use as an Artificial Vascular Graft in Cardiovascular Disease"

_materials, 2021, doi:10.3390/ma14051239_

Round 1
Reviewer 1 Report
Dear Authors,
the paper is well written and easy to understand. Here, point by point, some suggestions to improve the paper:
- I think there is an error with measure unit in line 103 where 15 nm diameter is reported.
- when you say that (line 272): "Manufacturing conditions for the ideal vascular graft should be easy and accurately reproducible". I agree. Since your method report a dip-coating process, please add a comment regarding the reproducibility of such process step and if it can affect the effectiveness of the proposed grafts.
- The cited literature seems a little bit out of date, especially as regards the 3D printing of medical devices. For example, the recent paper "Long-term results of triple-layered small diameter vascular grafts in sheep carotid arteries", reporting a process similar to the one reported in this paper is not cited.
- If I understand well, this paper is the natural follow on from the previous paper Lee, J.E.; Park, S.J.; Yoon, Y.; Son, Y.; Park, S.H. Fabrication of 3D freeform porous tubular constructs with mechanical flexibility mimicking that of soft vascular tissue. Journal of the mechanical behavior of biomedical materials 2019, 91, 193-201, doi:10.1016/j.jmbbm.2018.12.020. Also considering my previous comment, it seems that the novelty element is not strong enough. I would suggest a short comment on a more updated literature in the field of 3D printed vascular grafts, highlighting the novelty introduced in this work.
Author Response
Thank you for the valuable comments for our manuscript.
1) The typo of unit, “15 nm”, has been revised as “15 mm”.
2) As reviewer pointed out, the reproducibility is the important factor in our developed method. However, in this study, we focused on the feasibility test of the as-manufactured graft. We are working on the reproducibility test using our method in a separate study. Instead, we added a comment about the aspects of easy fabrication and shape customization as below.
[line 251] “Manufacturing conditions for the ideal vascular graft should be easy and accurately reproducible. Such grafts can be rapidly manufactured with diverse diameters and lengths as well as shapes, and easily stored and shipped. In this respect, our method including 3D printing and solution coating is characterized with facile manufacturing and shape customizability.”
3) As for the reference, we included the additional references including the reviewer’s suggestion.
39. Wang, C.; Li, Z.; Zhang, L.; Sun, W.; Zhou, J. Long-term results of triple-layered small diameter vascular grafts in sheep carotid arteries. Med Eng Phys 2020, 85, 1-6, doi:10.1016/j.medengphy.2020.09.007.
40. Kang, T.Y.; Lee, J.H.; Kim, B.J.; Kang, J.A.; Hong, J.M.; Kim, B.S.; Cha, H.J.; Rhie, J.W.; Cho, D.W. In vivo endothelization of tubular vascular grafts through in situ recruitment of endothelial and endothelial progenitor cells by RGD-fused mussel adhesive proteins. Biofabrication 2015, 7, 015007, doi:10.1088/1758-5090/7/1/015007.
4) Rather than on the novelty in fabrication method, this study focused on the possibility of clinical application of the artificial vascular graft fabricated by the 3D printing-based method reported previously. The previous paper did not include any demonstration about biological study. Thus, in this manuscript, in-vitro and in-vivo experiments were performed for confirming the feasibility of our previously developed grafts. We believed our results would suggest the potential for the clinical applications.
Reviewer 2 Report
This is a valuable contribution related to the new synthetic polymer vascular grafts for cardiovascular application. The authors show the biocompatibility and reduction of thrombogenesis for their 3D templated grafts in the early postoperative stages of vascular surgery compared with standard grafts, providing perspective for clinical applications in vascular surgeries The paper is well documented and written and there is no doubt from my side that it can be published in its present form.
Author Response
Thank you for the comments.
Reviewer 3 Report
Ji Eun Lee and coauthors suggested a new method of 3D-printing, dip coating, and salt leaching for soft tissue engineering. Thermoplastic polyurethane (TPU) was used as dip-coating material. This method was applied for fabrication of soft porous tubes which can be used for blood vessel replacement. In terms of mechanical properties the tubes made of TPU are better compatible with blood vessels in comparison with prostheses made of PTFE. The authors examined the histological changes in rat aortas that were surgically opened and sutured with PFTE or TPU material. It was demonstrated that when using TPU, thrombogenesis is less pronounced. For other indicators (neovascularization, calcification, inflammatory infiltrates, fibrosis) the changes were similar. Thus, the authors suggested that 3D-templated vascular grafts can be used as a substitute of PFTE prostheses in vascular surgery.
Comments.
he data shown in Fig. 5 should be described in more detail. The difference in the degree of endothelialization is not very noticeable.
Author Response
Thank you for your kind review. Your comments are very important and useful for this article. We all agree on your opinion. Therefore, we changed our Fig.5. We have added the sentences. “The endothelial cells are originally flat cells, but when activated they turn into rounds.”